# Systematic exploration of local reviews of the care of maternal deaths in the UK and Ireland between 2012 and 2014: a case note review study

Fiona Cross-Sudworth,[1] Marian Knight,[2] Laura Goodwin,[1] Sara Kenyon[1]

[1]Institute of Applied Health Research, University of Birmingham, Birmingham, UK
[2]Nuffield Department of Population Health, National Perinatal Epidemiology Unit, University of Oxford, Oxford, UK

**Correspondence to**
Sara Kenyon;
s.kenyon@bham.ac.uk

## ABSTRACT

**Objectives** Local reviews of the care of women who die in pregnancy and post-birth should be undertaken. We investigated the quantity and quality of hospital reviews.

**Design** Anonymised case notes review.

**Participants** All 233 women in the UK and Ireland who died during or up to 6 weeks after pregnancy from any cause related to or aggravated by pregnancy or its management in 2012–2014.

**Main outcome measures** The number of local reviews undertaken. Quality was assessed by the composition of the review panel, whether root causes were systematically assessed and actions detailed.

**Results** The care of 177/233 (76%) women who died was reviewed locally. The care of women who died in early pregnancy and after 28 days post-birth was less likely to be reviewed as was the care of women who died outside maternity services and who died from mental health-related causes. 140 local reviews were available for assessment. Multidisciplinary review was undertaken for 65% (91/140). External involvement in review occurred in 12% (17/140) and of the family in 14% (19/140). The root causes of deaths were systematically assessed according to national guidance in 13% (18/140). In 88% (123/140) actions were recommended to improve future care, with a timeline and person responsible identified in 55% (77/140). Audit to monitor implementation of changes was recommended in 14% (19/140).

**Conclusions** This systematic assessment of local reviews of care demonstrated that not all hospitals undertake a review of care of women who die during or after pregnancy and in the majority quality is lacking. The care of these women should be reviewed using a standardised robust process including root cause analysis to maximise learning and undertaken by an appropriate multidisciplinary team who are given training, support and adequate time.

## INTRODUCTION

For over six decades, the care of all women who die during or shortly after pregnancy in the UK has been independently reviewed through a process of confidential enquiries (CEs). These are an internationally acknowledged method of reviewing the care of

### Strengths and limitations of this study

► This is the first study to systematically examine the number and quality of local reviews of the care of women who died during or after pregnancy in the UK and Ireland.

► This study systematically examined the quantity and quality of local reviews of maternal deaths within the UK and Ireland over a 3-year period, which may not be representative of local reviews over a wider time period or in different countries.

► Each review was assessed on the basis of what was contained within the anonymised case notes provided to Mothers and Babies: Reducing Risk through Audits and Confidential Enquiries across the UK (MBRRACE) and therefore may not reflect the full procedure of review in some cases.

individuals who die or have severe complications in order to learn from adverse outcomes and reduce the incidence.[1 2] The principles have been utilised globally to review care of women who have died in pregnancy or in the postnatal period.[3] Countries or states that have utilised systematic CE methodology include France,[4] Sweden,[5] USA states of Washington[6] and California,[7] Tanzania,[8] Australia,[9] India[10] and South Africa.[11] However, it is less clear as to the quality of the review and it is hard to establish from the literature whether there is a standardised approach in individual countries as to the content of the review. Since 2012, a collaboration called Mothers and Babies: Reducing Risk through Audits and Confidential Enquiries across the UK (MBRRACE–UK) has been responsible for the continuation of the national programme of CEs and the surveillance of all perinatal and maternal deaths, as well as reviews of selected predetermined significant morbidities. These CEs use multidisciplinary teams (MDTs) of clinicians from outside the region where the woman's death occurred, to review anonymised case notes (medical records)

and assess the care given against national guidelines. Assessment is undertaken by these independent reviewers and a consensus regarding whether care was good or required improvement, and if so, whether these may have made a difference to the woman's outcome, is made at a multidisciplinary meeting. Findings from a maternal death CE published in 2018 identified that improvements in care may have made a difference to their outcome for 38% of all women who died.[12]

There has been some controversy over the impact of these reports. Some believe that that they have acted as a catalyst for significant improvements in maternity care across the UK and contributed towards the reduction in the national mortality rate.[13] Others, however, have questioned whether incident reporting systems such as national CE or audit are responsible for these improvements.[14] Indeed, it is argued that local review of adverse outcomes is needed, in addition to national data, in order to facilitate ownership of relevant issues and therefore increase the chance of change in practice.[15]

At local level in the UK and Ireland, maternal deaths ordinarily trigger a serious incident review from the hospital providing the majority of care or where the woman died. Maternal deaths are rare events and usually described as 'unexpected and avoidable' and as such considered under the 'Serious Incident Framework' (SIF),[16] wherever the death. SIFs contain the explicit recommendation that contributory factors and/or root causes should be examined to identify fundamental issues and ensure a full understanding of the event to maximise the learning opportunity. The focus is to consider system errors, rather than a review of individual clinicians. The National Patient Safety Agency developed a root cause analysis (RCA) investigation toolkit[17] which provides a structured way of examining nine potential contributory factors: patient, staff, task, communication, equipment, work environment, organisational, education and training and team factors. There is currently no systematic assessment of the quality of local hospital-based reviews of the care of women who die during or shortly after pregnancy. The aim of this study was to investigate the quantity and quality of local hospital reviews following maternal deaths (during pregnancy and up until 42 days after birth) between 2012 and 2014 inclusive using MBRRACE-UK anonymised case records.

## METHODS

All maternal deaths that occurred in the UK and Ireland in the 3-year period from 2012 to 2014 (inclusive) in early pregnancy and up to 42 days after birth were reviewed utilising anonymised case record review. Access to the anonymised case notes was via the MBRRACE-UK secure website and included case notes, statements and summaries as well as local review reports.

The objectives were to identify the proportion of hospital reviews carried out when there was a maternal death relating to the time, place and cause of death; to establish which professionals were involved in reviews; if the root causes had systematically been assessed as recommended by the SIF and whether there were resultant actions. If there were actions, whether they were individual or systematic and finally, to explore whether an audit was recommended to evaluate change in practice.

A data extraction form was developed to include key components of the SIF. For each local review, three authors independently undertook the data extraction and any differences between the data were resolved.

The major causes of antenatal and postnatal deaths were examined in relation to whether local reviews of care were undertaken. These were grouped according to whether the death was related to an obstetric, medical or psychiatric cause. Obstetric deaths were those due to amniotic fluid embolism, anaesthesia, deaths in early pregnancy, haemorrhage or eclampsia and pre-eclampsia. Deaths considered to be medical in cause included cardiac deaths, those due to malignancy, neurology, sepsis, thrombosis or thromboembolism and other medical causes.

The quality of each review was assessed based on the SIF by examining the composition of the review panel, whether a systematic examination of the root causes (contributory factors) was undertaken and whether any actions resulted, and audit was undertaken to evidence changes to practice. The composition of each review panel was examined and the profession of those involved as documented in the review.

### Patient and public involvement

This research was done without patient involvement as this study utilised anonymised case notes for secondary analysis.

## RESULTS

### Number and type of reviews of care of women who died

There were 262 maternal deaths that occurred between 2012 and 2014. Women who died from accidental or incidental causes such as road traffic accidents (n=24) were excluded and five sets of case notes were unavailable which resulted in 233 maternal death cases for assessment. Of the 233 maternal deaths, 177 (76%) were reviewed by the hospital where the majority of care had been given or where the woman had died. However, there was no evidence of a review having been undertaken in 56 deaths (24%) and no evidence of change in this proportion over time.

### Timing, place and cause of maternal deaths

The timing of maternal deaths was considered in relation to whether or not a review was completed. Of the 92 women who died in pregnancy, 45 (49%) of these occurred at less than 20 weeks gestation with 62% reviewed. After 20 weeks' gestation, a higher proportion of the deaths were reviewed (85%) (table 1). Of the 141 maternal deaths that occurred in the postnatal period, 78 (55%) occurred in the first week and of these 85% (n=66)

**Table 1** Timing (gestation and days) of women who died in pregnancy or in the early postnatal period

| Deaths of women (gestation or days) | | Reviewed Number (%) | Not reviewed Number (%) | Total |
|---|---|---|---|---|
| Deaths in antenatal period (gestation) | 0–20/40 | 28 (62) | 17 (38) | 45 |
| | 21–35/40 | 26 (87) | 4 (13) | 30 |
| | 36–42/40 | 14 (82) | 3 (18) | 17 |
| Antenatal total | | 68 (74) | 24 (26) | 92 |
| Deaths in postnatal period | 0–6 days | 66 (85) | 12 (15) | 78 |
| | 7–13 days | 15 (79) | 4 (21) | 19 |
| | 14–27 days | 19 (79) | 5 (21) | 24 |
| | 28–42 days | 9 (45) | 11 (55) | 20 |
| Postnatal total | | 109 (77) | 32 (23) | 141 |
| Total | | 177 (76%) | 56 (24%) | 233 |

were reviewed. Of the deaths between 28 and 42 days after birth, just under half (n=9, 45%) were reviewed.

### Place of death

An intensive care unit (ICU) was the most common location where women died (n=88); the care of 75% of the women who died in ICU was reviewed although maternity services were not always involved. Of the 30 women who died while being cared for in maternity services such as delivery suite, theatre and wards, 93% (n=28) were reviewed (table 2). Women who died in accident and emergency departments, however, were less likely to have their cases reviewed (28/42, 67%) along with those who died in specialist units such as neurological, liver, vascular or cardiac units or in palliative care (9/13, 69%).

### Causes of death

Of the major causes of maternal death, both obstetric and medical-related deaths had higher proportions of reviews compared with deaths related to psychiatric causes (table 3).

### The quality of the review

Of the women who died, 60% (n=140) had a documented local review of the care received. For a further

16% (n=37), a review was mentioned but this was not included in the case notes and so the quality could not be assessed, and 24% (n=56) had no review included in the case notes. The most common type of review was entitled a serious incident report (29%, n=68), with RCA being the title of 18% (n=41), hospital review of 12% (n=27) and 2% (n=4) having another title.

### Composition of review panels

Sixty-five per cent (91/140) of reviews were conducted by a MDT, although this did not always include maternity services, and 18% (25/140) were conducted by a single reviewer (table 4). Of the reviews undertaken, 60% (84/140) involved obstetricians or gynaecologists and 59% (82/140) included midwives. Absence of maternity service representation was evident in cases where the death occurred at a different hospital or non-maternity department of the same hospital (eg, accident and emergency or intensive care unit). For 16% (23/140) of reviews, the job title(s) of the professional(s) who undertook them was not documented. The family was documented as having specific questions or issues addressed by the panel in 14% (19/140) of reviews and external reviewers were involved in 12% (17/140) reviews.

The exact composition of the group was sometimes lacking and, while not explicitly recommended, the authors considered the optimum minimum composition of the MDT for review to be different for each of the three causes of death. Review of a maternal death from an obstetric cause should include an obstetrician, midwife and anaesthetist, and yet this was only documented in 12/30 (40%) of local reviews examined. Review of a maternal death from a medical cause should include an obstetrician, midwife and specialist in whatever the medical condition, such as a cardiac specialist, and yet this was documented in only 43/102 (42%) reviews examined. Review of a maternal death relating to a psychiatric cause should include an obstetrician, midwife and psychiatric specialist, and yet none of those examined did (0/8). Only 55/140 (39%) of maternal

**Table 2** Place of death

| Place of death | Reviewed Number (%) | Not reviewed Number (%) | Total |
|---|---|---|---|
| Accident and emergency | 28 (67) | 14 (33) | 42 |
| General hospital | 12 (80) | 3 (20) | 15 |
| Home | 26 (70) | 11 (30) | 37 |
| Intensive care unit | 66 (75) | 22 (25) | 88 |
| Maternity services | 28 (93) | 2 (9) | 30 |
| Outdoors | 8 (100) | 0 (0) | 8 |
| Specialist units | 9 (69) | 4 (31) | 13 |
| Total | 177 (76) | 56 (24) | 233 |

**Table 3** Cause of death

| Cause of death | Antenatal | | Postnatal | | |
| --- | --- | --- | --- | --- | --- |
| | Reviewed Number (%) | Not reviewed Number (%) | Reviewed Number (%) | Not reviewed Number (%) | Total (%) |
| Obstetric deaths | 12 (27) | 4 (9) | 24 (55) | 4 (9) | 44 (19) |
| Mental health-related deaths | 8 (38) | 4 (19) | 6 (29) | 3 (14) | 21 (9) |
| Medical deaths | 48 (28) | 16 (10) | 79 (47) | 25 (15) | 168 (72) |
| Total | 68 (29) | 24 (10) | 109 (47) | 32 (14) | 233 (100) |

deaths were considered to have been reviewed by an optimum MDT, with the composition being unclear for 71/140 (51%).

### Contributory factors

Contributory factors were systematically assessed in only 13% (18/140) of local reviews using the headings listed in national guidance (see table 5). A further 11% (15/140) used some of these factors: overall the most frequently reported factor was communication (31/140, 22%). A small proportion of reviews (4%) utilised headings to assess care which differed to those listed in national guidance, such as individual knowledge and skill, clinical, external, other factors, documentation or systems. In 32% (45/140) of reviews, contributory factors were described in a summary paragraph only and there was no evidence that contributory factors had been examined in 36% (50/140) of local reviews examined.

The majority of local reviews examined (88%) included actions to improve ongoing care; most of which were systemic (79%). None of the reviews reported individual actions alone, while 9% (12/140) included both systemic and individual actions. A small number of reviews (9/140) only included non-clinical actions such as conducting the review, completing death notifications or supporting hospital staff. Only 14% (19/140) of all reviews of the care of women who died recommended or undertook an audit to monitor implementation of changes.

**Table 4** Professional group of reviewers

| Professional group of reviewers | Total, n=140 (%) |
| --- | --- |
| Obstetrics/gynaecology | 84 (60) |
| Midwifery | 82 (59) |
| Anaesthetics | 41 (29) |
| Senior management | 48 (34) |
| Risk/governance | 69 (49) |
| Pathologist | 4 (3) |
| External | 17 (12) |
| Family | 19 (14) |
| Other professional(s) | 70 (50) |
| Not documented | 23 (16) |

## DISCUSSION

This is the first study to systematically examine the number and quality of local reviews of the care of women who died during or after pregnancy in the UK and Ireland. It shows that only three-quarters of maternal deaths are reviewed and has highlighted that the care of women who died at less than 20 weeks gestation and between 28 and 42 days after birth was less likely to be reviewed. The care of women who died outside maternity services (eg, at home) and women who died from mental health-related causes was also less likely to be reviewed. The study also suggests that a substantial proportion of these local reviews of care were not optimal, in that they were not undertaken by a multidisciplinary group, did not include RCA and made relatively weak recommendations and actions.

This study systematically examined the quantity and quality of local reviews of maternal deaths within the UK and Ireland over a 3-year period. As such, this may not be representative of local reviews over a wider time period or in different countries. Assessment was made on the basis of what was contained within the anonymised case notes provided to MBRRACE-UK, and therefore may not reflect the full procedure of review in some cases.

**Table 5** Inclusion of contributory factors and follow-up in root cause analysis

| Root cause analysis content | Number, n=140 (%) |
| --- | --- |
| All individual contributory factors listed | 18 (13) |
| Some factors using National Patient Safety Agency headings | 15 (11) |
| Some factors using different headings | 5 (4) |
| Mixed headings | 7 (5) |
| Summary only | 45 (32) |
| No contributory factors | 50 (35) |
| Actions (or recommendations/learning points) | 123 (88) |
| No actions | 17 (12) |
| Systemic actions | 111 (79) |
| Systemic and individual actions | 12 (9) |
| Non-clinical actions only | 9 (6) |
| Timeline and person responsible identified | 77 (55) |
| Audit | 19 (14) |

The study findings appear to be consistent within a wider context of reviews of care related to both maternal morbidity and perinatal death in maternity services. Shah *et al*[18] examined severe maternal morbidity reviews from six UK hospitals and identified that the care of some women who had severe morbidities was not reviewed and in those that were, key issues affecting the outcome were not always identified, or were lessons evidenced as being learnt. A comparison of American local and statewide reviews of 31 maternal deaths found that state reviews found more preventable system rather than patient factors when the cases were anonymised and investigated by an external review team.[19] National CEs into the care of women who had term, normally formed antepartum stillbirths found that 23% had a local review carried out and only 10% were undertaken according to Royal College of Obstetrics and Gynaecology guidance.[20] While some of these CEs were not focused on maternal deaths, it appears that there is a lack of effective investigation of care within a hospital after a significant adverse outcome.

There is increasing evidence that the use of RCA within healthcare is problematic with variable use of the investigation tools in reviews of serious incidents.[21] This is further compounded by the complexity of reviews being undertaken within tight deadlines and by a local team who may have provided care or work alongside those who have, which may reduce objectivity. Indeed, this may explain why such reviews commonly result in weak corrective actions and poor dissemination of findings and that repetition of similar events continues[22 23] suggesting that lessons were not being learnt and that action to address issues was inadequate. There also appears to be tension between a 'no blame' culture and a 'just culture'[24] with the emotional tensions felt by those undertaking the review of care. A balance needs to be maintained between system and individual accountability; reviews should not be a scapegoat exercise while any professional failure must focus on learning and quality improvement. Suggested solutions to support quality balanced reviews include the need for professionalisation of incident investigation (including appropriate training), a need for the involvement of patient's relatives to be recognised and valued[25] and for a better understanding of the role of blame.[24] It is clear that the quality of reviews should be properly monitored and evaluated by an enhanced surveillance system, such as those not only in the UK but also elsewhere in Europe,[26 27] North America[28] and Australasia.[29]

The purpose of review is to learn from events and this should involve reporting, investigation, learning and action planning, implementation and closure[15] and yet of the reviews examined, not all had action plans, and just over half had a nominated person responsible for the action, with audit only recommended to check change in practice in 14%. While not systematically assessed, many of the recommendations were for guidelines to be updated, training to be undertaken or dissemination of the findings. There is some evidence that easily achieved actions do not work as effectively as system changes but these are most commonly found in reviews, due to the comparatively lower financial and time costs, as well as the reduced pressure to change the culture of organisations.[20] These 'weaker' types of actions may not prevent the event from happening again.[30] Further research is needed to explore the follow-up to local review including the short-term and long-term impact of actions.

## CONCLUSIONS

This study is the first to show that not all women who die in pregnancy or in the first 42 days post-birth in the UK and Ireland have their care reviewed by the local hospital. It also identified variation across hospitals concerning who was involved in reviews as well as the quality.

The care of all women who die during or after pregnancy needs to be reviewed using a standardised robust process by an appropriate MDT. If care that can be improved is identified through the review, strong actions that will change practice are necessary with audit to monitor implementation to improve practice.

Hospitals need to allocate sufficient time for preparation, participation and appropriate follow-up for the review of care. Training is required for those involved in reviews to ensure adequate assessment of maternity service systems, culture and care, not just at the time of death, in order to come to a clear understanding of the mother's care and what if anything, could be done to prevent the same outcome happening again.

**Acknowledgements** We would like to acknowledge the contribution of the many healthcare professionals and staff from the health service and other organisations who were involved in the notification of cases, the provision of data and the assessment of individual cases in both the UK and Ireland.

**Contributors** The study was designed by SK and MK. Data extraction was undertaken by FCS, LG and SK. FCS wrote the first draft of the paper which was commented on by all authors, who have approved the submitted version. All authors had access to the data in the study and can take responsibility for the integrity of the data and the accuracy of the data analysis.

**Funding** This paper presents independent research arising from a Research Professorship award to Professor Marian Knight, NIHR-RP-011-032, supported by the National Institute for Health Research. The views expressed in this publication are those of the author(s) and not necessarily those of the NHS, the National Institute for Health Research or the Department of Health. This work was also supported by the Collaborations for Leadership in Applied Health Research and Care in the West Midlands (Grant number IS-CLA-0113-10018).

**Competing interests** None declared.

**Patient consent for publication** Not required.

**Ethics approval** Permission was obtained from the Healthcare Quality Improvement Partnership as data controllers to enable access to the anonymised case notes for this secondary case note review study. Research Ethics Committee approval was not required for this secondary analysis of anonymous data.

**Provenance and peer review** Not commissioned; externally peer reviewed.

**Data sharing statement** Data are available for sharing on request from HQIP at https://www.hqip.org.uk/national-programmes/accessingncapop-data/#.XAFiOi10do4.

and indication of whether changes were made. See: https://creativecommons.org/licenses/by/4.0/.

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
