## [Reviewer comments · BMJ Open]

ARTICLE DETAILS

TITLE (PROVISIONAL)	Systematic exploration of local reviews of the care of maternal deaths in the UK and Ireland between 2012- 2014: a case note review study
AUTHORS	Cross-Sudworth, Fiona; Knight, Marian; Goodwin, Laura; Kenyon, Sara

VERSION 1 - REVIEW

REVIEWER	Christine H. Morton Research sociologist California Maternal Quality Care Collaborative Stanford University School of Medicine Palo Alto, CA USA
REVIEW RETURNED	24-Feb-2019

GENERAL COMMENTS	Thank you for this important work to examine number and quality of local reviews of maternal deaths. The paper is well written, mostly clear and worthy of publication. I have a few comments and requests for clarity regarding some of the methods and the discussion. Page 3 line 14-15; consider adding California USA to this list, California Department of Public Health. (2018). The California Pregnancy-Associated Mortality Review. Report from 2002 - 2007 maternal death reviews. Retrieved from https://www.cdph.ca.gov/Programs/CFH/DMCAH/CDPH%20Document%20Library/PAMR/CA-PAMR-Report-1.pdf line 29-30 "Assessment is undertaken..." Do the reviewers meet together to come to consensus? How are the independent reviews brought together? Who does that? Page 4 Line 43-47. Please clarify whether the 'authors' are the same as the 'investigators'? What are the background/training of the authors and/or investigators? Page 7 Lines 27-29 Please clarify the first sentence - "...had a documented review on the care received contained within the medical records..." I don't quite understand what this means. Also are 'notes' the same as 'medical records'? how do 'case notes' relate to these two? Line 34 I see an extra comma. n=68,)
---

	Page 10 Researchers in Illinois USA compared statewide and regional reviews of maternal death. They found “The statewide MMRC found more potential preventability and determined that preventability was associated with provider and systems factors, not patient factors. Observed discrepancies between regional perinatal center and statewide MMRC reviews were likely due to the complexity of cases selected for review, the multidisciplinary external composition of the review team, and the de-identification of cases. Multidisciplinary statewide expert panels should be implemented in addition to local and regionalized reviews.” Geller, SE et al Maternal & Child Health Journal. Dec2015, Vol. 19 Issue 12, p2621-2626. You may find this interesting Lines 29-30 Please clarify what is meant by a “local team who may not be independent” –independent from whom? Implications? Lines 35-38. Expand on your observation of an apparent tension between ‘no blame’ and ‘just’ cultures ... emotional tensions... professional hierarchies. How did you see this in your data? Or do these observations arise from the authors’ professional experiences? Line 38-39 “solutions identified”. By whom? How do these solutions track back to the data presented in the paper?
--	--

REVIEWER	Serena Donati National Institute of Health, Italy
REVIEW RETURNED	20-Mar-2019

GENERAL COMMENTS	The paper is very interesting for those who manage a maternal mortality surveillance system because it highlights the critical issues related to the review process of maternal death cases. It is clear that the quality of the reviews deserves improvement and that this aspect should be properly monitored and evaluated by an enhanced surveillance system . On the other hand the paper could be too specialized for the general public, the descriptive analyzes perhaps a bit poor and the inferability of the results, as reported by the authors, limited. I advise the authors to shorten and simplify the title of the paper.
---

VERSION 1 – AUTHOR RESPONSE

Reviewer Comments	Changes made	Line number
Reviewer 1: Thank you for this important work to examine number and quality of local reviews of maternal deaths. The paper is well written, mostly clear and worthy of publication. I have a few comments and requests for clarity regarding some of the methods and the discussion.	Thank you. We welcome the comments to improve the paper.	-
Page 3	Reference added	47

line 14-15; consider adding California USA to this list, California Department of Public Health. (2018). The California Pregnancy-Associated Mortality Review. Report from 2002 - 2007 maternal death reviews. Retrieved from https://www.cdph.ca.gov/Programs/CFH/DMC/AH/CDPH%20Document%20Library/PAMR/C-A-PAMR-Report-1.pdf		
line 29-30 “Assessment is undertaken....” Do the reviewers meet together to come to consensus? How are the independent reviews brought together? Who does that?	Added to sentence for clarity: These CEs use multi-disciplinary teams of clinicians from outside the region where the woman’s death occurred, to review anonymised case notes (medical records) and assess the care given against national guidelines. Assessment is undertaken by these independent reviewers and a consensus regarding whether care was good or improvements were noted, and if so, whether these may have made a difference to the woman’s outcome is made at a multi-disciplinary meeting.	53-7
Page 4 Line 43-47. Please clarify whether the ‘authors’ are the same as the ‘investigators’? What are the background/training of the authors and/or investigators?	Yes authors are the investigators and ‘between investigators’ has been removed to reduce confusion. Two authors are midwives and researchers and were the primary assessors of the case notes and reviews. The other author involved in the assessments is a researcher. All three collaboratively worked together to utilise the proforma in a standardised manner.	97
Also are ‘notes’ the same as ‘medical records’? how do ‘case notes’ relate to these two?	The first reference to case notes ‘medical records’ has been inserted in brackets to add clarity to the case notes term. Notes, case notes and medical records have been used interchangeably – these terms have been changed where needed, to all state ‘case notes’.	9, 55, 158, 160
Line 34 I see an extra comma. n=68,),	Removed	162
Page 7 Lines 27-29 Please clarify the first sentence - ...”had a documented review on the care received contained within the medical records....” I don’t quite understand what this means.	Revised sentence to say: Of the women who died, 60% (n=140) had a documented local review on the care received (have removed the end of the sentence).	158-9

Page 10 Researchers in Illinois USA compared statewide and regional reviews of maternal death. They found “The statewide MMRC found more potential preventability and determined that preventability was associated with provider and systems factors, not patient factors. Observed discrepancies between regional perinatal center and statewide MMRC reviews were likely due to the complexity of cases selected for review, the multidisciplinary external composition of the review team, and the de-identification of cases. Multidisciplinary statewide expert panels should be implemented in addition to local and regionalized reviews.” Geller, SE et al Maternal & Child Health Journal. Dec2015, Vol. 19 Issue 12, p2621-2626. You may find this interesting	Added following sentence into the Discussion: A comparison of American local and statewide reviews of 31 maternal deaths found that state reviews found more preventable system rather than patient factors when the cases were anonymised and investigated by an external review team.	227-230
Lines 29-30 Please clarify what is meant by a “local team who may not be independent” – independent from whom? Implications?	Removed ‘not be independent’ and added instead: ...have provided care or work alongside those who have, which may reduce objectivity.	238
Lines 35-38. Expand on your observation of an apparent tension between ‘no blame’ and ‘just’ cultures ... emotional tensions... professional hierarchies. How did you see this in your data? Or do these observations arise from the authors’ professional experiences?	Tension was not seen in the data but arises from observations from professional experience and some literature e.g. Peerally et al, 2017. Added: A balance needs to be maintained between system and individual accountability; reviews should not be a scapegoat exercise while any professional failure must focus on learning and quality improvement.	244
Line 38-39 “solutions identified”. By whom? How do these solutions track back to the data presented in the paper?	Suggestions that have already been made have had a reference added. Added for clarity: Suggested solutions to support quality balanced reviews include the need for professionalisation...	248, 249 246
Reviewer: 2 The paper is very interesting for those who manage a maternal mortality surveillance system because it highlights the critical issues related to the review process of maternal death cases. It is clear that the quality of the reviews deserves improvement and that this aspect should be properly monitored and	Thank you, we agree with the reviewer comment and have added this to the manuscript.	249-251

evaluated by an enhanced surveillance system.		
On the other hand the paper could be too specialized for the general public, the descriptive analyzes perhaps a bit poor and the inferability of the results, as reported by the authors, limited.	We agree the paper is too specialised for the general public. We have therefore specifically aimed the paper at healthcare professionals who understand the concept of case review even if they have not been involved in them, and grouping within a BMJ Open specific topic area will emphasise this. The analysis has been strengthened by the suggested comments from the reviewers and made more generally applicable by reference to additional data from the US, as well as existing maternal mortality surveillance systems. It supports existing evidence that local reviews are often not consistent or robust and do not prevent reoccurrence. While there are always limitations to studies, we consider that this study's limitations do not negate the impact of the findings and the potential transferability of the results. We believe that additional highlighting of the role of enhanced surveillance systems, as the reviewer notes above, strengthens the vitally important message of this article in supporting a rising awareness of the need to improve learning from critical incidents.	-
I advise the authors to shorten and simplify the title of the paper.	The title has been shortened to: Systematic exploration of local reviews of the care of maternal deaths in the UK and Ireland between 2012- 2014: a case note review study	1-2